# Space-efficient optical computing with an integrated chip diffractive neural network

H. H. Zhu[1], J. Zou[1], H. Zhang[1], Y. Z. Shi[2], S. B. Luo[1], N. Wang[3], H. Cai[3], L. X. Wan[1], B. Wang[1], X. D. Jiang [1✉], J. Thompson [4], X. S. Luo [5], X. H. Zhou [6✉], L. M. Xiao [7✉], W. Huang[8], L. Patrick[5], M. Gu [9✉], L. C. Kwek [1,4✉] & A. Q. Liu [1✉]

Large-scale, highly integrated and low-power-consuming hardware is becoming progressively more important for realizing optical neural networks (ONNs) capable of advanced optical computing. Traditional experimental implementations need $N^2$ units such as Mach-Zehnder interferometers (MZIs) for an input dimension $N$ to realize typical computing operations (convolutions and matrix multiplication), resulting in limited scalability and consuming excessive power. Here, we propose the integrated diffractive optical network for implementing parallel Fourier transforms, convolution operations and application-specific optical computing using two ultracompact diffractive cells (Fourier transform operation) and only $N$ MZIs. The footprint and energy consumption scales linearly with the input data dimension, instead of the quadratic scaling in the traditional ONN framework. A ~10-fold reduction in both footprint and energy consumption, as well as equal high accuracy with previous MZI-based ONNs was experimentally achieved for computations performed on the *MNIST* and *Fashion-MNIST* datasets. The integrated diffractive optical network (IDNN) chip demonstrates a promising avenue towards scalable and low-power-consumption optical computational chips for optical-artificial-intelligence.

[1] Quantum Science and Engineering Centre (QSec), Nanyang Technological University, Singapore 639798, Singapore. [2] National Key Laboratory of Science and Technology on Micro/Nano Fabrication, Department of Micro/Nano Electronics, Shanghai Jiao Tong University, Shanghai 200240, China. [3] Institute of Microelectronics, A*STAR (Agency for Science, Technology and Research), Singapore 138634, Singapore. [4] Centre for Quantum Technologies, National University of Singapore, Singapore 117543, Singapore. [5] Advanced Micro Foundry, 11 Science Park Road, 117685 Singapore, Singapore. [6] State Key Joint Laboratory of ESPC, Center for Sensor Technology of Environment and Health, School of Environment, Tsinghua University, Beijing 100084, China. [7] Shanghai Engineering Research Center of Ultra-Precision Optical Manufacturing, School of Information Science and Technology, Fudan University, Shanghai 200433, China. [8] Suzhou Institute of Nano-Tech and Nano-Bionics (SINANO), Chinese Academy of Sciences (CAS), Suzhou 215123, China. [9] Quantum Hub, School of Physical and Mathematical Science, Nanyang Technological University, 50 Nanyang Ave, 639798 Singapore, Singapore. ✉email: exdjiang@ntu.edu.sg; xhzhou@mail.tsinghua.edu.cn; liminxiao@fudan.edu.cn; gumile@ntu.edu.sg; cqtklc@nus.edu.sg; eaqliu@ntu.edu.sg

Optical neural networks (ONNs) that exploit photonic hardware acceleration to compute complex matrix-vector multiplication, have the advantages of ultra-high bandwidth, high calculation speed, and high parallelism over electronic counterparts[1,2]. With the rapidly increasing complexity in data manipulation techniques and dataset size, highly integrated and scalable ONN hardware with ultracompact size and the reduced energy consumption is strongly desired to conquer the resource bottleneck in artificial neural networks. Due to the advantages of ultracompact size, high-density integration, and power-efficiency silicon photonic integrated circuits (PICs) are emerging as a promising candidate for establishing large and compact computing units predominantly required in optical-artificial-intelligence computers[3–5]. Moreover, the fabrication of silicon PICs is compatible with CMOS technology, thus current infrastructure can support mass production at a low cost[6]. Typical silicon PIC architectures to realize chip-scale ONNs[7–15] use cascades of multiple Mach–Zehnder interferometers (MZIs)[7–13] and microring resonator-based wavelength division multiplexing technology[14,15]. Though PICs have shown great potential in integrated optics, the space utilization (directional couplers and phase modulators)[16], energy consumption (heaters to control phase), and the complex control circuits for reducing fluctuation of the resonance wavelength[17,18] restrict the development of a large-scale programmable photonic neural network. Additionally, phase-change materials such as $Ge_2Sb_2Te_5$[5], inverse-designed metastructures[19], and nanoscale neural medium[20] combined with CMOS-compatible integrated photonic platform, can further minimize the size of integrated photonic circuits. However, the number of units for these ONN architectures scales quadratically with the input data dimension[5,7,14] for both convolution operations and matrix multiplication, leading to quadratic scaling in the footprint and energy consumption.

Meanwhile, Fourier transforms and convolutions, which are the fundamental building blocks of ONN architectures, have been realized using various approaches including the spatial light modulators[21–24], micro-lens array[25–28], and holographic elements[29–33]. Fourier transforms and convolutions as well as applications in classification tasks have been proposed utilizing the optical diffraction masks in free space[26,31]. However, these free-space approaches require heavy auxiliary or modulation equipment which is space-consuming[34,35] and exceedingly challenging to program in real-time[31,36,37], hindering their feasibilities towards the large-scale implementation and commercialization of photonic neural networks.

Here, we demonstrate the first scalable integrated diffractive neural network (IDNN) chip using silicon PICs, which is capable of performing the parallel Fourier transform and convolution operations. Due to the utilization of on-chip compact diffractive cells (slab waveguides), both the footprint and power consumption of the proposed architecture is reduced from quadratic scaling in the input data dimensions required for MZI-based ONN architectures[7,10,13] to linear scaling for the IDNN. This reduction in the resource scaling from quadratic to linear will have a profound impact on the realization of large-scale silicon-photonics computing circuits with current fabrication technologies[38–40].

The parallel Fourier transform and convolution operations in our IDNN chip can be applied to classification tasks with Iris flower, Handwriting digit, and Fashion product datasets. The cases of 1D sequence and 2D digit image recognition demonstrate the enhanced performance of the convolution operations. Two typical cases of Iris flower classification with a one-layer neural network, and Handwriting digit and Fashion product classification with a two-layer neural network as well as the comparison with the fully connected neural network, are conducted to further characterize the classification performance of our IDNN chip. For the *Iris* dataset classification, we obtain a chip testing accuracy of 98.3% in the one-layer IDNN using a complex modulation. For Handwriting and Fashion recognition tasks using 500 different sample inputs, the testing accuracies of 89.3 and 81.3% are obtained, respectively, which are equivalent to those obtained in a fully connected neural network. Compared to the previous design with an area of 5 mm$^2$ and power of 3–7 W[7,10,13], our IDNN chip achieves the reduced hardware footprint (0.53 mm$^2$) and low power consumption (17.5 mW), manifesting the advantage of the Fourier-based design as a scalable and power-efficient solution for data-heavy artificial intelligence applications.

## Results

**Design and fabrication.** A typical electronic one-layer neural network consists of an input layer, an output layer, connections between them (weight matrix), and elementwise nonlinearity (activation function). In a departure from the electronic neural network, our introduced IDNN framework implements a convolution transformation physically in the optical field using PICs. The convolution transformation is special matrix multiplication, and the complex-valued matrix elements are circulant[41]. The theoretical framework of the IDNN with multi-layers is shown in Fig. 1a. The preprocessed signals are encoded via modulating the amplitude and phase (two degrees of freedom) of coherent light and then input to the multi-layer IDNNs. In each layer as depicted in Fig. 1b, the information propagates through matrix multiplications followed by a nonlinear activation function. The linear part of each layer is composed of two ultracompact diffractive cells to implement optical discrete Fourier transform (ODFT) operation ($W^l_{ODFT}$) and optical inverse discrete Fourier transform (OIDFT) operation ($W^l_{OIDFT}$), and a complex-valued transmission modulation region in the Fourier domain behind the ODFT operation to achieve the Hadamard product. The nonlinear activation function $f$ originates from the intensity detection of the complex outputs. We can hereby achieve a single-layer neural network. This IDNN can also be multi-layered by cascading multiple chips or recycling one single chip. The recycling process can be realized by configuring the phase shifters through a computer-controlled digital-to-analog converter.

Figure 1c shows the holistic diagram of the proposed silicon PICs-based IDNN chip, which consists of four major sections, e.g., input preparation, ODFT operation, complex modulation, and OIDFT operation. A continuous-wave laser (wavelength 1550 nm) is incoupled into the device through a grating coupler to realize the input source. The basic computing unit of a modulated MZI cell comprises multimode interferometers (MMI)-based beam splitters and two thermo-optic phase shifters in the "input preparation" section. The internal phase shifters between two beam splitters (indicated as $\theta$) and external beam splitters (indicated as $\phi$) are used to control the amplitude and phase of the input signals, respectively. All phase shifters are thermally tuned by the TiN heaters that are packaged to a printed circuit board (Fig. S5a). Then, they can be manipulated by an external electric circuit controller.

Subsequently, the ten input signals are fed into the diffractive cell, which is composed of a slab waveguide to implement the Fourier transform in the "ODFT operation" section (details will be shown in Fig. 2). After the operation, an array of MZIs is utilized to modulate the amplitude and phase of the ten signals after the ODFT operation. The modulated signals are then introduced to another diffractive cell to implement the OIDFT. After that, the ten output intensity signals are detected by an array of photodetectors. In addition, a thermoelectric controller is implemented as the substrate beneath the chip to control and

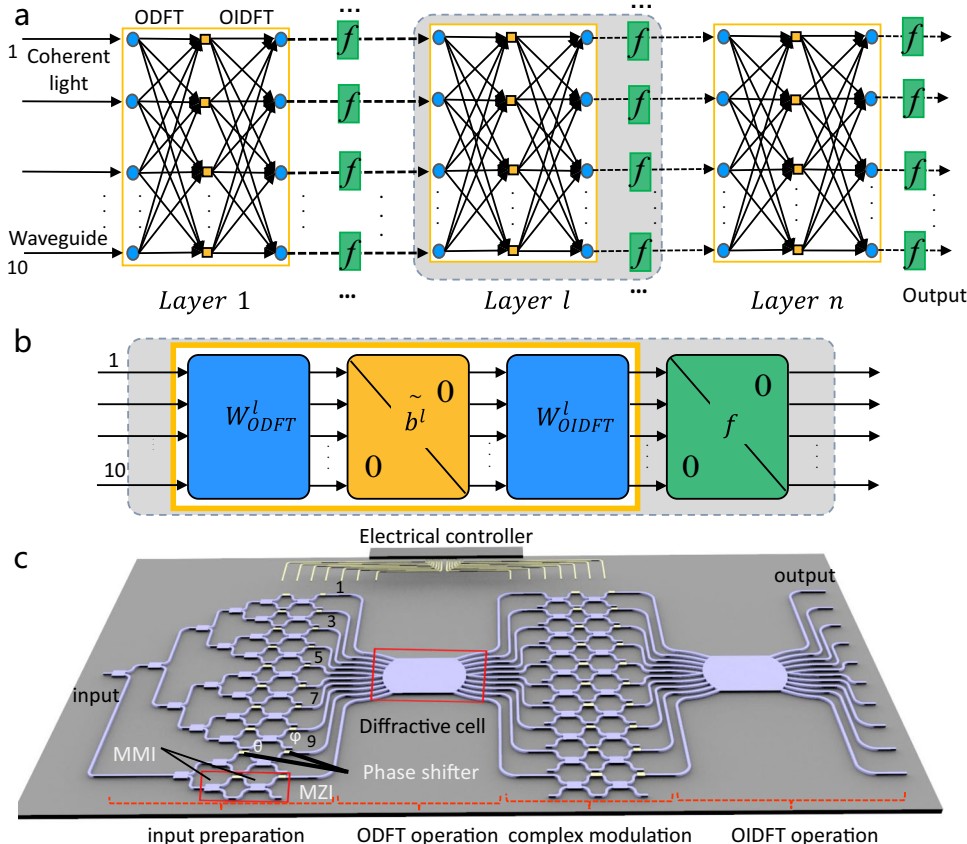

**Fig. 1 Optical integrated diffractive neural networks (IDNNs). a** The multi-layer neural networks. One layer contains three main parts: optical discrete Fourier transform (ODFT) operation, amplitude/phase modulation, and optical inverse discrete Fourier transform (OIDFT) operation. A nonlinear activation function is added between two layers. **b** IDNN operates on complex-valued inputs using coherent light. There are two matrices based on diffractive cells and a Hadamard product operation raised by phase and amplitude modulation behind the ODFT operation. **c** Schematics of the experimental device. The device includes four functional parts: (1) input signal preparation; (2) implementing ODFT operation; (3) modulating amplitude/phase in the Fourier domain; (4) implementing OIDFT operation.

stabilize the temperature (details in Supplementary Note 2). The holistic 3.2 mm × 2 mm chip monolithically integrates 10 modes, 20 MZIs, 2 slab waveguides, and 40 thermo-optic phase shifters.

Figure 2a, b shows the optical micrographs of the whole chip and the diffractive cell, respectively. The Fourier transform, as a core operation of our network, can be achieved using a slab waveguide-based ultracompact diffraction cell, whose input waveguides are coded by different phase shifters incorporated in the "input preparation" section. The normalized electric field after the ODFT operation [$E_i(k)$] (Supplementary Note 3) can be expressed as

$$E_i(k) = \frac{1}{j\sqrt{\lambda_0 R/n_s}} \int E_0(x) \exp(-j2\pi f_k x) dx \qquad (1)$$

where $E_0$ is the input electric field, $f_k = kn_s/(\lambda_0 R)$ is the angle spectrum of the incident light field, $\lambda_0$ is the wavelength of light in vacuum, $R$ is the arc radius of the diffraction cell as shown in Fig. 2c, $n_s$ is the effective refractive index of the waveguide, and $k$ is the axis of output coordinate plane.

The simulated electric field distribution in the diffraction region and the output intensity distribution is shown in Fig. 2c, d, respectively. The input light is centered into two central waveguides along the propagation direction and the outputs are the electric field profile after the ODFT operation. The detailed analysis for light propagating in the entire chip is shown in Supplementary Note 3. The measured normalized intensities of ODFT operation in the diffractive cell and the retrieved signal

from all output channels after OIDFT operation are shown in Fig. 2e, f, respectively. The input signal can be well retrieved after ODFT and OIDFT operations are applied with no intervening phase modulation as seen from Fig. 2f (more experimental results can be found in Fig. S7).

**Image recognition with correlation algorithm.** To demonstrate the performance of our IDNN chip and validate the design for realizing the convolution matrix, we first use the chip as a convolution operator to implement image recognition using the correlation algorithm. The calibration for the initial amplitude and phase of the signal, the realization of ODFT and OIDFT operations, and the implementation of weight matrices are shown in Supplementary Note 4. The detailed mathematical elucidation of correlation algorithms can be found in Supplementary Note 5.

The 1D and 2D convolution operations have various applications in machine learning, such as convolution layers in convolutional neural networks[41–43] and correlation recognition of human faces from still images or video frames[44,45]. Here, we demonstrate the cases of 1D sequence and 2D image recognitions using a classical correlation algorithm[44], which demonstrates the capability of the convolution operation realized by our IDNN chip. The image recognition process is shown in Fig. 3a. The correlation calculation of the input test image ($n \times n$) and target image ($m \times m$) is conducted using the Hadamard product between the Fourier transform of the input test image and the Fourier transform of a target image. Then the height of that peak

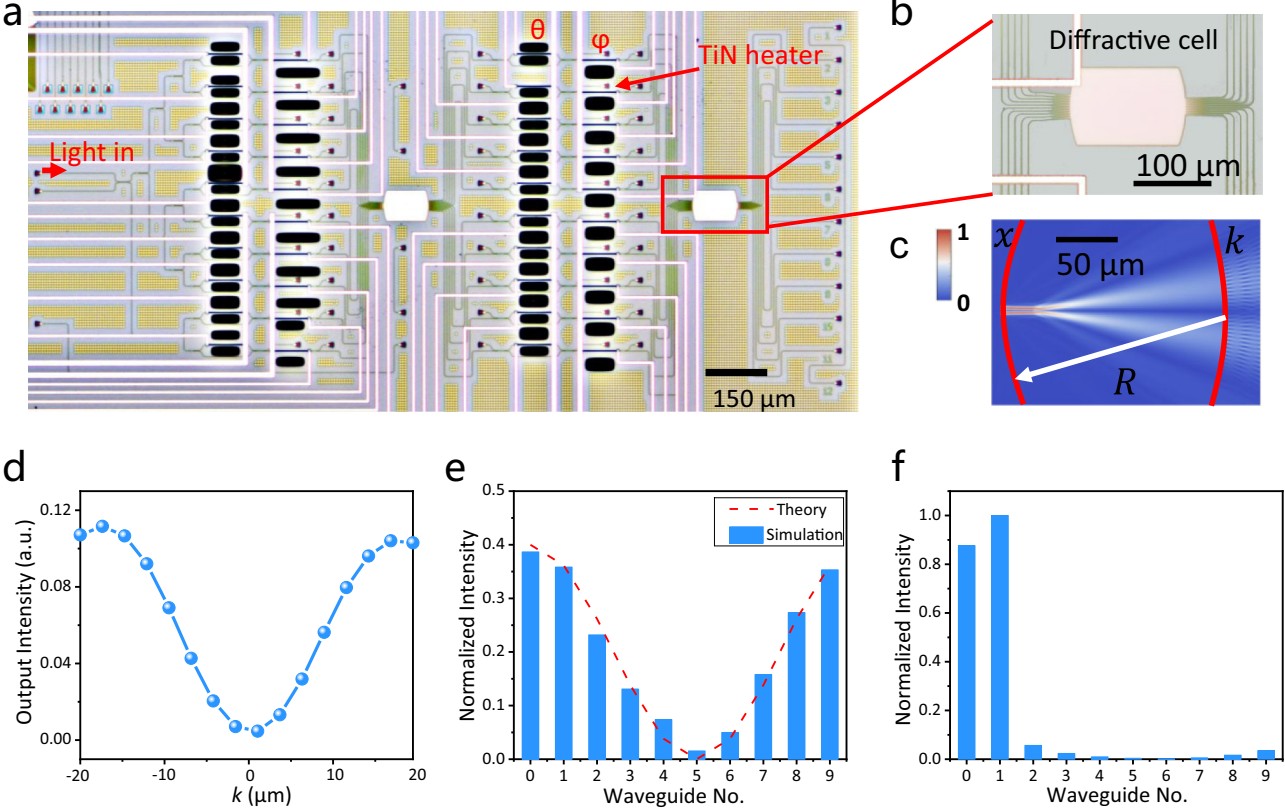

**Fig. 2 Optical microscope photos and some simulation results. a** The optical micrograph of the whole chip. **b** The optical micrograph of the diffractive cell. **c** Simulated electric field distribution in the slab waveguide region. **d** Simulated output amplitude distribution along the output waveguide plane. **e** Simulated normalized intensity of ten channels after ODFT operation. **f** Simulated normalized retrieved intensity information of ten channels.

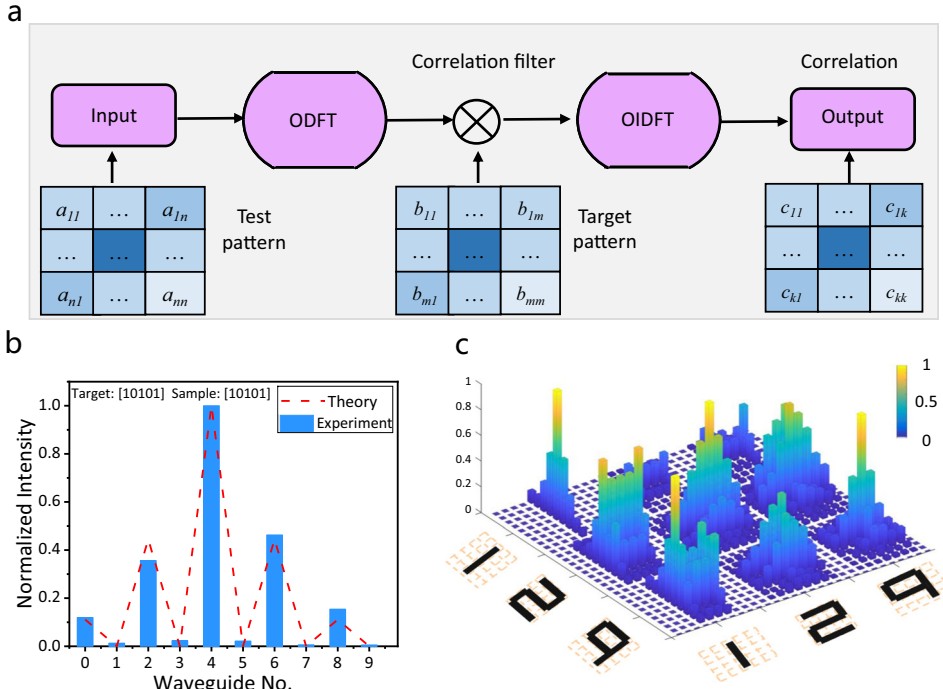

**Fig. 3 Image recognition. a** Schematics of the image recognition process. The input test image dimension is $n \times n$, target image dimension is $m \times m$ and the output dimension is $k = n + m - 1$. **b** Experimental normalized intensity from the output waveguides when the sample sequence [10101] is the same as the target sequence [10101]. **c** Experimental correlation results for digit image recognition. There are three-digit images (1, 2, 9), and each digit image is composed of $5 \times 5$ matrix.

(relative to the background) of the output result is used to determine whether the test image matches the target image.

The experimental recognized sequence ([10101]) is the same as the target sequence as shown in Fig. 3b (More results in Fig. S8). Furthermore, we extend the 1D sequence to 2D image recognition of digits ($5 \times 5$ matrices) with the same correlation algorithm. The correlation signals for nine different digit-image combinations are shown in Fig. 3c (details in Supplemental Note 6). Strong correlation peaks are observed along the diagonal line of the measured cross-correlation matrix, reflecting the intuition that two images have maximum correlation and similarity when they are the same. Our approach to convolutional processing provides an effective method for accelerating computation via reducing computational complexity, as compared with traditional methods[43,41]. The traditional electronic single convolution operation involves $(n - m + 1)^2$ matrix-vector multiplication (MVM) operations with $n \times n$ and $m \times m$ dimension matrices[43,41]. Similarly, the state-of-the-art research works for photonic convolutional accelerators[5,35] also perform convolution operation by MVM operations, but in combination with parallel operations using WDM technology to accelerate the computing. Here, we realize an optical Fourier transform-based convolution. Each 2D convolution with $n \times n$ and $m \times m$ dimension matrices needs $(n - m + 1) \times m$ 1D convolution operations[46]. Since a 1D convolution is converted into an MVM operation in our design, our approach only needs $(n - m + 1) \times m$ MVM operations to compute a 2D convolution. This design can thus accelerate the convolution operations to achieve improved scaling over traditional electronic counterparts[43,41]. More discussions on the computational convolution approach can be found in Supplementary Note 6. It can further be extended to circulant convolution operation[45] and employed in typical classification tasks[47–49].

**Iris flower classifier**. We demonstrate the performance of our network in classifying the Iris flower dataset. Here we have four input parameters (the lengths and widths of the sepals and petals of a candidate flower). The task is to determine which of the three possible subspecies the flower belongs to (*setosa, versicolor, and virginica*). We implement this classification task using a one-layer neural network with eight neurons as shown in Fig. 4a. The input and output are connected by a complex, circulant weight matrix (*W*) with eight trained complex-valued parameters. Input signals are encoded by modulating the amplitude of the input field of the IDNN. Amplitude and phase modulators in the Fourier domain are trained to map inputs into three output waveguides, which are detected by external power sensors. The highest intensity measurement outcome is used to indicate the subspecies.

The entire dataset with 150 instances is split into the training set and testing set according to a ratio of 8:2. The weights are trained only on the training set. In the training process, the amplitude and phase of each channel in the Fourier domain are learnable parameters, representing a complex-valued modulation on the neural network. Trained with the error back-propagation algorithm[16], the numerical convergence of the accuracy and loss versus epoch number are depicted in Fig. S10. The training accuracy of 98.0% is achieved in the complex modulation, while the training accuracy is 97.3% in the conventional one-layer fully connected neural network (Fig. S10).

After training, the performance of the IDNN chip with complex modulation is evaluated on the Iris flower classification testing dataset, see Fig. 4. The experimentally obtained energy distribution from the eight output waveguides is shown in Fig. 4b for one testing data point as an example. The channel "0" has the highest output intensity, indicating that this flower is classified as "Setosa". More experimental results can be found in Fig. S10. The

intensity distribution from the output detectors reveals that the IDNN can achieve a maximum signal at the corresponding detector channel identified with candidate flower species. About 150 samples, including the training set and testing set, are extracted into two features and visualized as a scatter plot in Fig. 4c. A total of 30 testing samples are experimentally evaluated using the IDNN chip with only one wrong classification as marked by the red circle in Fig. 4c. The one-layer IDNN chip can achieve a high classification accuracy of 96.7% over 30 testing images in the experiment (Fig. 4d), which is the same as the testing result (96.7%) of the conventional one-layer fully connected neural network, manifesting that our IDNN has comparable accuracy with a fully connected network. However, our design has fewer components (i.e., the number of MZIs is reduced from 16 to 8) and more efficient space overhead compared with the fully connected architectures.

**Handwritten digit and Fashion product classifier**. More complicated datasets, *MNIST* (handwrite digits images), and *Fashion-MNIST* (clothing images) datasets are used to further validate the functionality of our IDNN chip. The two datasets are both split into the training (60,000 images) and testing sets (10,000 images). Our model is trained on the entire training set, and 500 instances are uniformly and randomly drawn from the testing set to validate the trained model on-chip[35]. Figure 5a shows that the network is composed of two layers, a hidden layer $W$ with $16 \times 10$ trained complex parameters and an output layer $W^{out}$ with ten trained complex parameters. The outputs of the output layer are the recognition results (The computing details are shown in Supplementary Note 8). The numerical testing accuracy and loss vary with the epoch number as shown in Fig. 5b with a classification accuracy of 92.5% (see Fig. S11). Besides, the simulated energy distribution from the intensity detection results of ten outputs is shown in Fig. S11f to display the classification performance, indicating that our numerical simulations predict the channel with the highest energy will generally correspond to the correct handwritten digit. We experimentally tested 500 images and the confusion matrix (Fig. 5c) shows an accuracy of 91.4% in the generated predictions, in contrast to 92.6% in the numerical simulation. The output energy distributions from 10 waveguides for the classification of the digit "2" are depicted in Fig. 5d (More experimental results are shown in Fig. S12). These results show that we have successfully implemented the classification task on the integrated diffractive optical computing platform. Our experimental testing accuracy is slightly lower (91.4%) than the numerically predicted value obtained from simulations (92.6%), which is attributed to the errors from the diffractive region, heater calibration, and other experimental factors (Supplementary Note 10). Although our method does not achieve as high accuracy as previous reservoir computing demonstrations[50,51], their internal dynamics of reservoirs are uncontrollable with fixed nodes, which leads to the stringent requirement of obtaining many parameters with high accuracy. Consequently, they are difficult to be implemented on chip. In contrast, our approach requires fewer parameters in controllable weight matrices that significantly reduce the footprint and energy consumption. Meanwhile, the input data compression process in our method also loses some information and lowers the numerical classification accuracy. The numerical blind testing accuracy (92.5%) with 10,000 samples can be further improved after enlarging the size of the network with more input information. It is worth noting that the classification accuracy obtained with our approach is comparable to the fully connected network (Fig. 5b), but our method consumes less energy and occupies less footprint, representing the major advantages of the design.

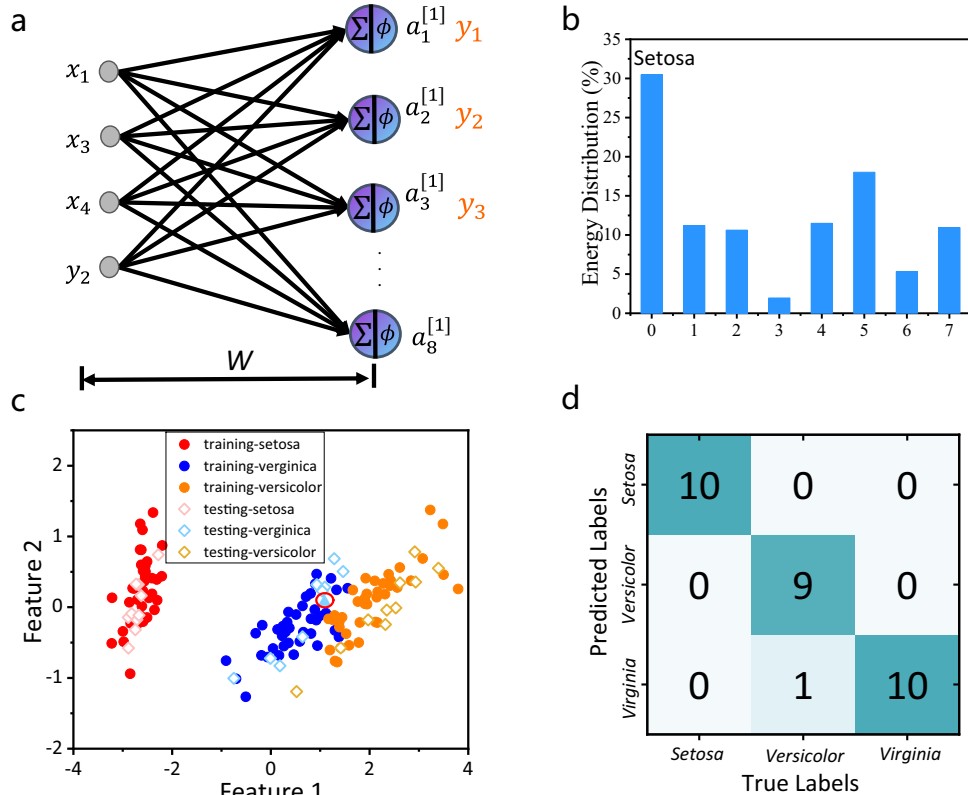

**Fig. 4 Iris flower classification using the diffractive neural network. a** Network consists of one layer. The outputs of the layer are the recognition results after the intensity detection. $x_1$ is sepal length, $x_2$ is sepal width, $x_3$ is petal length, and $x_4$ is petal width. $y_1$ is *Setosa*, $y_2$ is *Versicolor*, and $y_3$ is *Verginica*. **b** Experimental output intensity distribution of the IDNN for a class of flowers as "*Setosa*" is demonstrated. **c** Training and testing results of classification. One classification error appears on the testing part. **d** Confusion matrix in our experiment using 30 different flowers.

We further use the IDNN to perform the classification of the *Fashion-MNIST* dataset. In general, we can encode the preprocessed signals by using the amplitude and phase channels of coherent light. In the case of the *MNIST* dataset classifier, the output of the first hidden layer is encoded to the amplitude channels of the output layer. To verify that the network also performs well when input signals are encoded into the phase channel, the signal from the input layer of the fashion product images is encoded in the phase channel in our experiment. The numerical testing results for the classification of fashion products are depicted in Fig. S13. The corresponding accuracy for 500 testing images with one hidden layer is 82.0% in simulation versus 80.2% in an experiment (Fig. 5f). The experimental energy distributions from ten waveguide outputs using one image of a "pullover", is shown in Fig. 5g and the experimental results from other images are depicted in Fig. S14. We implement additional experiments using the phase channel encoding input signal on the *MNIST* dataset and using the amplitude channel encoding input signal on the *Fashion-MNIST* dataset. The confusion matrix for 500 images (Fig. S15a) shows an experimental accuracy of 89.4%, in contrast to 92.6% for the numerical simulation results on the *MNIST* dataset. The results for the *Fashion-MNIST* dataset are 81.4% according to the numerical simulations versus 80.4% for the experiment (Fig. S15b). We note that the accuracy error between the experimental and numerical results has no distinct difference between using the amplitude channel and the phase channel. However, for phase encoding, the intensity of the input signal would not be attenuated, which leads to a higher output intensity in our experiment.

We compare our IDNN architecture with a fully connected neural network with the training results of the *MNIST* dataset and *Fashion-MNIST* dataset as shown in Fig. 5b, e, respectively. The

simulation model is built in Pytorch with a learning rate of 0.0001, a training period of 500 iterations, and a batch size of 100. Our complex circulant matrix-based neural network with one hidden layer achieves a numerical testing accuracy of 92.5%, which is comparable to that obtained in both a fully connected neural network with one hidden layer (93.4%) and a circulant matrix-based neural network with three-layers (93.5%) for the *MNIST* dataset. For the *Fashion-MNIST* dataset, both the simulated testing accuracies of the complex circulant matrix-based neural network with one hidden layer and three layers are higher than 81% (81.7 and 83.2%, respectively), which is comparable with 83.0% for fully connected based algorithm. For the *MNIST* and *Fashion-MNIST* datasets, the classification accuracy of our IDNN chip with less trainable parameters can be comparable with the traditional fully connected network.

Our IDNN chip can achieve comparable testing accuracy with classical fully connected neural networks in datasets *MNIST* and *Fashion-MNIST*. Some key metrics evaluating the physical network are listed by the comparison with the traditional fully connected MZI-based[7,10,13] architectures for the optical realization, as shown in Table 1. For the *MNIST* and *Fashion-MNIST* datasets, a $10 \times 10$ matrix on-chip is achieved with only 0.53 mm$^2$ area and 17.5 mW power consumption to maintain the phase of heaters, while the traditional MZI-based ONN architectures need an area around 5 mm$^2$ and requisite energy consumption of 3–7 W. (More details are in Supplementary Notes 1).

## Discussion

We propose and experimentally demonstrate the reconfigurable IDNN chip, leveraging diffractive optics to realize an efficient

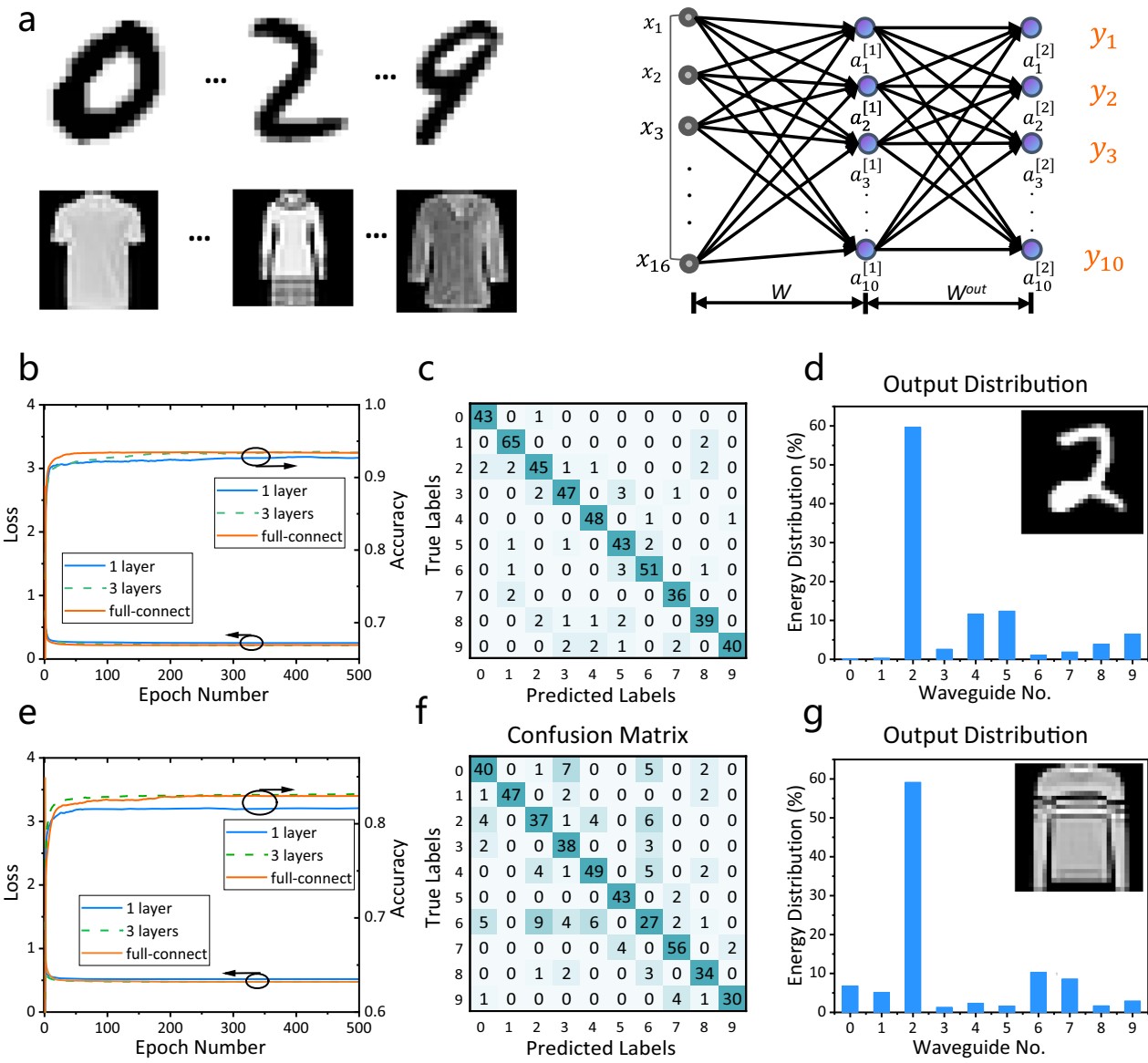

**Fig. 5 Handwriting and fashion recognition using the IDNN. a** The network consists of a hidden layer $W$ and an output layer $W^{out}$. The outputs of the output layer are recognition results. The input layer is calculated by the traditional computer and the complex output results of the input layer are converted into amplitude and phase information as the input of the chip. **b** The numerical testing results of accuracy and loss versus epoch number for the *MNIST* dataset. **c** The confusion matrix for our experimental results, using 500 different handwritten digits. **d** The output intensity distribution of the IDNN for a handwritten input of "2" is demonstrated. **e** The numerical testing results of accuracy and loss versus epoch number for the *MNIST-Fashion* dataset. **f** The confusion matrix in the experiment. **g** As an example, the output intensity distribution of the IDNN for a fashion product input of "pullover" is demonstrated.

**Table 1 The performance of our proposed IDNN framework.**

| Type | The no. modulators | Matrix dimension | Area (10 × 10 neurons) | Power consumption (10 × 10 neurons) |
|---|---|---|---|---|
| MZI-VMM[10] | 16 | 4 × 4 | 5.2 mm$^2$ | ------ |
| MZI-VMM[7] | 9 | 4 × 4 | 6 mm$^2$ | 3 W |
| MZI-VMM[13] | 15 | 4 × 4 | 4.8 mm$^2$ | 7 W |
| Our work | 10 | 10 × 10 | 0.53 mm$^2$ | 17.5 mW |

matrix-vector multiplication. Previous MZI-based integrated photonic approaches for computing have been predominantly restricted by large footprints (around 0.5 mm$^2$ per MZI unit[23,26,29]) and the excessive energy consumption to tune the phase, e.g., one needs ~50 mW to tune each thermo-optic heater ($2\pi$ phase shift). Our approach can effectively decrease footprints and energy

consumption (see Table 1) by reducing the numbers of MZI from quadratic scaling with input data size to linear. The key strategy of the scaling reduction is using integrated ultracompact diffractive cells (slab waveguides) to replace MZI units and realize Fourier transform operations. The slab waveguide is a passive device with a small size (0.15 mm$^2$) and does not require electric tuning.

Our IDNN chip shows an equivalent classification capability while cutting down the number of optical basic components (i.e., MZI modulators) to 10% of original on *MNIST* and *Fashion-MNIST* classification tasks and a half of original on the *Iris* datasets, compared with the previous MZI-based ONN architectures. We achieved a $10 \times 10$ matrix on-chip with 10 MZIs and two diffractive cells, which has an area of $0.53 \, \text{mm}^2$, while 100 MZIs are required with an area around $5 \, \text{mm}^2$ for a fully connected network. Meanwhile, the power consumption used to maintain the phase of modulators, which is also the dominant source of energy consumption, decreases from 3–7 W (previous techniques) to 17.5 mW (see Fig. S1). Using the compact design method, the matrix size can easily be scaled up to $64 \times 64$ with acceptable loss (see Supplementary Note 1). Considering the current fabrication level of the integrated photonic chip, reduction in resource scaling from quadratic to linear and the relevant footprint and energy consumption reduction are profoundly meaningful towards the goal of a large-scale programmable photonic neural network and the achievement of photonic AI computing. It is worth mentioning that the IDNN focuses on implementing the convolution matrix with the advantages of scalability and low power consumption, instead of an arbitrary matrix-vector multiplication modulation. Since a convolution matrix contains fewer free parameters than those in an arbitrary matrix, the current demonstrated one-layer network may not be an optimal solution for complicated classification tasks. This issue can be mitigated by implementing multi-layers or multiple convolutions in each layer to obtain a stronger expression capability. The speed of IDNN is also limited by the electrical equipment and can be accelerated when the modulators and detectors are high-speed programmed.

Our integrated chip is adaptable to the Fourier-based convolution acceleration algorithm and multiple tasks, which has great potentials for many applications in compact and scalable application-specific optical computing, future optical-artificial-intelligence computers, and quantum information processing, such as image analysis, object classification, and live video processing in autonomous driving[5,42]. The footprint and energy consumption scale linearly with the input data dimension, which will hugely reduce the resources used for future sophisticated and data-heavy computing, and facilitate the prospects of a power-efficient, ultracompact, and large-scale integrated optical computing chip.

## Methods

**Fabrication**. The whole optical diffractive neural network is fabricated on the silicon-on-insulator (SOI) platform with a 220-nm thick silicon top layer and a 2-μm thick buried oxide. Subsequently, a thin layer of titanium nitride (TiN) is deposited as the resistive layer for heaters. A thin aluminum film is patterned as the electrical connection to the heaters and photodetectors. Isolation trenches are created by etching the SiO₂ top cladding and Si substrate.

**ODFT operation**. The critical part of the IDNN chip is the ODFT/OIDFT operations composed of the diffractive regions with a suitable phase difference from different waveguides before the diffractive components. The phase shift, $\phi_{n-k}$, between input $k$ and output $n$ can be designed to satisfy the relation $\phi_{n-k} = \frac{2\pi}{N}(k - \frac{N-1}{2})(n - \frac{N-1}{2})$ by setting positions of input and output waveguides. In addition, the phase offset $\varphi_k = \frac{\pi}{N}(N-1)k$ is additionally added to the input $k$ using length adjustment or a phase shift. Then the phase difference $\Delta\phi_n$ between two paths to the output $n$ originating from the inputs, $k+1$ and $k$ is derived as $\Delta\phi_n = (\phi_{n-(k+1)} + \varphi_{k+1}) - (\phi_{n-k} + \varphi_k) = \frac{2\pi n}{N}$. This allows the ODFT computation to be realized.

**Circulant matrix**. For the linear part of the IDNN, a complex circulant matrix multiplication is achieved utilizing ODFT and OIDFT operations. Assuming each circulant matrix $B$ is defined by a vector $b$, which is the first-row vector of $B$. Based on the theories of circulant convolution[52–54] and diffractive optics, the matrix-vector multiply can be efficiently performed using Fourier transform, which is expressed as $y^{l+1} = W^l_{OIDFT}[(W^l_{ODFT} \circ y^l) \circ \tilde{b}^l] = F^{-1}[F(y^l) \circ F(b^l)]$. $W^l_{ODFT}$ is the

matrix for ODFT, $W^l_{OIDFT}$ is the matrix for OIDFT, $b^l$ is a vector of the circulant matrix $B$ of the $l$-layer neural network, $y^l$ is the input vector of the $l$-layer neural network. $F(\circ)$ represents an $n$-point ODFT operation, $F^{-1}(\circ)$ is the inverse of $F(\circ)$ (OIDFT operation), $\circ$ represents the Hadamard product and $\tilde{b}^l$ is the Fourier transform of $b^l$.

**Numerical simulation**. The training process of Iris flower, Handwritten digit, and Fashion product datasets was conducted in Pytorch (Google Inc.), a package for Python. The training process of Iris flower, Handwritten digit, and Fashion product datasets was conducted in Pytorch (Google Inc.), a package of Python. The learned complex parameter ($w$) is expressed as $w = a + ib = Ae^{i\alpha}$ where $A$ is encoded into the internal phase shifters ($\theta$) with $\theta = \arcsin(A^2)$ and $\alpha$ is encoded into the external phase shifters ($\phi$) with $\phi = \alpha - \theta/2$. The ODFT and OIDFT operations are implemented using the angular spectrum method. A three-dimensional finite-difference-time-domain method is used to simulate the optical field distribution and transmission spectra of the diffractive cells.

## Data availability
The data that support the findings of this study are available from the corresponding authors on reasonable request.

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

## Acknowledgements

This work was supported by the Singapore National Research Foundation under the Competitive Research Program (NRFCRP13-2014-01) and the Singapore Ministry of Education (MOE) Tier 3 grant (MOE2017-T3-1-001).

## Author contributions

H.H.Z., L.C.K., M.G., and A.Q.L. jointly conceived the idea. H.H.Z., J.Z., H.Z., S.B.L., X.D.J., and A.Q.L. performed the numerical simulations and theoretical analysis. H.H.Z., J.Z., and L.X.W. did the experiments. H.H.Z., B.W., N.W., H.C., Y.Z.S., X.S.L., L.P., X.D.J., L.C.K., M.G., and A.Q.L. involved in the discussion and data analysis. H.H.Z., Y.Z.S., J.T., M.G., and A.Q.L. prepared the manuscript. X.H.Z., L.M.X., and W.H. revised the manuscript. L.C.K., M.G., and A.Q.L. supervised and coordinated all the work. All authors commented on the manuscript.

## Competing interests

The authors declare no competing interests.
