## [Peer Review File · Nature Communications]

This manuscript demonstrates an integrated diffractive optical network (IDNN) to achieve parallel Fourier transforms and convolution operations. Traditional optical neural networks (ONN) require N^2 units for an N -dimensional input, which leads to quadratic scalings in both the footprint and the power consumption. By virtue of the on-chip compact diffractive cells in the proposed IDNN framework, such quadratic scaling is reduced to a linear one, which may shed a light on the realization of large-scale silicon-photonics computing circuits. Specifically, the authors concretely carry out the following tasks to reveal the enhanced performance of parallel Fourier transforms and convolution operations in IDNN:

1. Correlation pattern recognitions of 1D sequences and 2D digit images.
2. Classification tasks with Iris flower, MNIST handwriting digit and Fashion-MNIST product datasets.

The topic of this work is of great interest for both the experimental optics and the quantum machine learning community. The experimental results are sound, and the numerical effort is also noticeable. Hence I would recommend this manuscript for Nature Communications, after addressing the following major comments/suggestions:

1. I suggest the authors composing one paragraph in the supplementary information to mathematically elucidate the correlation pattern recognition, as well as the Hadamard product operation raised by phase and amplitude modulation, which would make the manuscript more readable.
2. Line 207: "60 samples (30 from the testing set and 30 from the sampling set) are experimentally tested ...". I do not understand why 30 additional samples are added into the test set. I guess the simulated accuracy (97.3%) of the traditional one-layer fully connected neural network is still evaluated from the original test set with 30 samples (a ratio of 8:2). It seems like the comparison between IDNN and the fully connected neural network is slightly unfair, although their accuracy are comparable.

Except for the above major concerns, I also have some minor points for improving the presentation:

1. Acronyms should be expanded in the abstract, IDNN \rightarrow integrated diffractive optical network.
2. The meaning of the subscripts of E in Eq.(1), Eq.(S3), Eq.(S5) and Eq.(S6) should be specified and consistent.
3. There are 10 channels in the chip, however, the digit number of the input sequences in the caption of Fig.S9(b)(c)(d) is 11.
4. Line 119: There is no Fig.S3a.

To conclude, I would recommend the publication of this manuscript if the issues raised above are substantially addressed.

Reviewer #2 (Remarks to the Author):

The manuscript "Space-Efficient Optical Computing and An Integrated Chip of Diffractive Neural Network" presents a design for an optical neural network with a small footprint and reduced energy consumption. The authors evaluate the performance of this design for machine learning classification tasks in experiments and numerical simulations.

The results presented in this manuscript show that this design has the potential to be more scalable than previous approaches to implement integrated optical neural networks. However, there are several aspects of this work that need further clarification:

- 1) The presented design has a clear advantage in scalability and power consumption compared to previous approaches. However, the manuscript lacks a critical discussion on the limitations of this approach. It should be noted that not all problems that can be solved with optical neural networks can be tackled with this approach.
- 2) The authors mention that the training/learning of this photonic system for the machine learning classification tasks has been done using Pytorch. However, there is no mention to the number of parameters that are trained. There is also no mention to how the learned parameters are transferred from the numerical simulations to the experiments. For instance, it would be interesting to know if the training has been informed with data coming from the experiments. The difficulty of a proper training of physical systems is a crucial aspect that appears to be disregarded in the current work.
- 3) Along the main text, numerical and experimental results are reported. The separation between what has been estimated from numerical simulations and the results that have actually been obtained from experiments is often unclear. A more precise writing would benefit the manuscript.
- 4) The manuscript contains several bold statements that I feel are too detached from reality. For instance, the sentence "...facilitating the realization of hundreds of millions of neurons" feels unrealistic given that here 10 input modes have been demonstrated. Similarly, the sentence "... can find tremendous applications..." should be more precise in terms of the applications that such a design can perform.
- 5) The performance for the classification tasks is relatively poor in the experimental evaluations. In particular, for the MNIST database, the authors report an accuracy of 89.3% in the experiments for 150 tested images. There are two aspects here that need further explanations. First, the authors should indicate how the 150 tested images in the experiment were extracted from the dataset of 10000 testing images. This is particularly relevant since some digits of the MNIST database are significantly easier to distinguish than others. Second, the performance for the MNIST of previous

works in optical neural networks reach higher accuracies, close to 99%. See e.g. M. Hermans et al. *Journal of Machine Learning Research* 16 (2015) 2081-2097 or P. Antonik et al. *IEEE Journal of Selected Topics in Quantum Electronics* 26 (2020) 7700812. I wonder if the reduced performance here is due to the small size of the optical network or for the difficulty to transfer the training parameters from the numerical simulations to the experiments (please comment if there are other relevant factors).

6) The supplementary information is quite extensive and contains as many as 18 figures. In my opinion, it is unclear what is shown in several of these figures. I find that crucial descriptions and explanations are missing at several points of the text. Although this criticism affects mainly to the supplementary information, it partly affects to the results presented in the main text, which would also benefit from a more streamlined description of the results.

Reviewer #3 (Remarks to the Author):

This work proposes an on-chip generalized diffractive neural network in Fourier-space to achieve the photonic convolution and fully connected neural network architectures. The proposed architecture (integrated diffractive neural network, IDNN) applies the slab star coupler (see reference [6] in supplementary material), i.e., the diffractive cell, to implement the Fourier transform, and adopts the programmable Mach-Zehnder interferometer array as the diffractive modulation layer. Implementing the $N \times N$ convolution matrix, the IDNN allows achieving a smaller computing area and higher integration density (~ 10 -fold reduction, see Table 1) compared with the previous MZI-based vector-matrix multiplication, which implies the potential application of the proposed approach for large-scale photonic computing. Different applications are experimentally demonstrated to validate the effectiveness of the proposed approach. The manuscript is well written and organized. However, the following concerns need to be addressed before I can make the recommendation.

* The discussion section should make it clear that the proposed approach cannot implement the arbitrary matrix-vector multiplication but the convolution matrix. In the MZI-based ONN [17], the authors decomposed the learnable matrix in each neural network layer via SVD, and then implemented matrix multiplication with SU(4) core and DMMC core of performing unitary matrix multiplication and diagonal matrix multiplication, respectively. While in this proposed IDNN, the authors replaced the unitary matrix multiplication operations with two Fourier transforms via diffraction. The unitary matrix multiplication in [17] is learned from data, while the Fourier transform in IDNN is fixed. It is not surprising that substituting two learnable components with a fixed Fourier transform can reduce the footprint and energy consumption.

* The experiment is conducted only on a small proportion of the testing dataset. For example, only 150 out of 10,000 different handwritten digits are tested in the MNIST dataset classification. What

are the reasons that hinder testing all the digits in the testing set since both the MZI and detector can be high-speed programmed and reconfigured?

* In Fig. 3, the digits of 4 and 9 are highly similar, resulting in the large peak values in the non-diagonal line. I would suggest using the different digits to make the statement in the last paragraph of page 8 more valid.

* The manuscript compares the optical convolution networks and electronic fully-connected networks to prove the performance of their suggested structure. I also suggest the comparisons and discussions with the electronic convolution network as well as the state-of-the-art research work of photonic convolutional accelerators, e.g., (1) Xu, X., et al. "11 TeraFLOPs per second photonic convolutional accelerator for deep learning optical neural networks." Nature 2021; and (2) Feldmann, J. , et al. "Parallel convolutional processing using an integrated photonic tensor core." Nature 2021. Implementing the convolution in Fourier-space is more efficient for the larger kernel size but not for the small kernel size often used in CNN.

* The authors attribute the larger error between the experimental and the simulation results on MNIST than on FASHION MNIST to the error of amplitude modulation. Can the authors add a phase-only modulation experiment on MNIST to prove this suspect? Or will the exclusion of amplitude modulation lead to obvious accuracy loss on MNIST?

Manuscript ID:	Nature Communications manuscript NCOMMS-21-17390
Paper title:	Space-Efficient Optical Computing and An Integrated Chip of Diffractive Neural Network
Authors:	H. H. Zhu, J. Zou, H. Zhang, Y. Z. Shi, S. B. Luo, N. Wang, H. Cai, L. X. Wan, B. Wang, X. D. Jiang, J. Thompson, X. S. Luo, X. H. Zhou, L.M. Xiao, W. Huang, L Patrick, M. Gu, L. C. Kwek, and A. Q. Liu

Reply to Reviewer 1

We are grateful to the Reviewer for the constructive comments, and the recognition of our contribution for the experimental optics machine learning community. We are happy to address the comments raised by the Reviewer. We thank the Reviewer for stimulating those improvements.

Comment 1: *I suggest the authors composing one paragraph in the supplementary information to mathematically elucidate the correlation pattern recognition, as well as the Hadamard product operation raised by phase and amplitude modulation, which would make the manuscript more readable.*

Answer 1: As per the reviewer’s suggestion, the mathematical elucidation of correlation pattern recognition and Hadamard product operation have been added in **Supplementary Note 5: Mathematical elucidation of correlation algorithms.**

“Supplementary Note 5: Mathematical elucidation of correlation algorithms.

In the first task of pattern recognitions of 1D sequences and 2D digit images, the goal is to obtain the similarity between two sequences or images. The cross-correlation, which is a commonly used metric for the similarity between two series, is represented by the symbol $R_{f,g}[n]$ as

$$R_{f,g}[n] = \text{cross - correlation}(f[j], g[j]) \quad (\text{S11})$$

where f and g are two series and n and j are the position numbers of the discrete sequence. The mathematical calculation of the correlation is the same as the convolution, except that the signal is not reversed before the multiplication process. Then, the relation between correlation and convolution is expressed as

$$R_{f,g}[n] = f[n] * g[-n] \quad (\text{S12})$$

where $*$ represents the convolution. The convolution is further converted into the point-wise multiplication in the Fourier transform region, and rewritten as

$$R_{f,g}[n] = \text{IDFT}\{\text{DFT}\{f[n]\} \circ \text{DFT}\{g[-n]\}\} \quad (\text{S13})$$

where $\text{DFT}\{f\}$ and $\text{DFT}\{g\}$ are the discrete Fourier transforms of $f[n]$ and $g[-n]$, respectively. \circ is the Hadamard product, which is also known as the element multiplication. By setting two vectors $U = [u_1, u_2, \dots, u_n]$ and $V = [v_1, v_2, \dots, v_n]$, the Hadamard product of the two vectors can be expressed as

$$W = U \circ V = [u_1 v_1, u_2 v_2, \dots, u_n v_n] \quad (\text{S14})$$

Finally, the correlation is converted to calculate the Hadamard product between the Fourier transforms of the input series f and the target series g . Since the Fourier transform converts real numbers to complex numbers, the Hadamard product in our operation is for complex numbers. In our chip, the complex numbers are easily encoded into the amplitude and phase components with the MZI modulators instead of real and imaginary parts. According to the transfer function (**Eq. (S10)**), the output field after the MZI modulator can be expressed as

$$E_{out} = Ae^{i\alpha} E_{in} = ie^{i(\phi + \frac{\theta}{2})} \sin(\frac{\theta}{2}) E_{in} \quad (\text{S15})$$

where the amplitude component is encoded into θ and phase component is encoded into $\phi + \frac{\theta}{2}$.

As a result, the correlation can be achieved by modulating the MZI using phase and amplitude modulation to evaluate the similarity between two series.”

The relevant description of this part is discussed in the revised manuscript as

“The calibration for the initial amplitude and phase of the signal, the realization of ODFT and OIDFT operations, and the implementation of weight matrices are shown in **Supplementary Note 4**. The detailed mathematical elucidation of correlation algorithms can be found in **Supplementary Note 5**.” in Line 8 on Page 8.

Comment 2: Line 207: "60 samples (30 from the testing set and 30 from the sampling set) are experimentally tested ...". I do not understand why 30 additional samples are added into the test set. I guess the simulated accuracy (97.3%) of the traditional one-layer fully connected neural network is still evaluated from the original test set with 30 samples (a ratio of 8). It seems like the comparison between IDNN and the fully connected neural network is slightly unfair, although their accuracy are comparable.

Answer 2: As pointed out by the reviewer, we realize that the adding of 30 additional samples from the training set into the testing set is indeed unreasonable, thus we revised this part and only used the experimental results collected from the 30 samples drawn from the testing set. The training accuracy of our design and the traditional one-layer fully connected neural network are 98.0% and 97.3%, respectively, as shown in **Fig. S12**. The blind testing accuracies with 30

samples of our design and the traditional one-layer fully connected neural network are the same, i.e., 96.7%. This segment of the text is revised in the revised manuscript as

“Trained with the error back-propagation algorithm [16], the numerical convergence of the accuracy and loss versus epoch number are depicted in **Fig. S10**. The **training** accuracy of 98.0% is achieved in the complex modulation, while the **training** accuracy is 97.3% in the conventional one-layer fully connected neural network (**Fig. S10**).” in Line 4 on Page 10 and

“A total of 30 testing samples are experimentally evaluated using the IDNN chip with only one wrong classification as marked by the red circle in **Fig. 4c**. The one-layer IDNN chip can achieve a high classification accuracy of 96.7% over 30 testing images in the experiment (**Fig. 4d**), which is the same as the testing result (96.7%) of the conventional one-layer fully connected neural network, manifesting that our IDNN has comparable accuracy with a fully connected network.” in Line 21 on Page 10.

Figure 4 has also been revised as

Fig. 4 Iris flower classification using the diffractive neural network. **a** Network consists of one layer. The outputs of the layer are the recognition results after the intensity detection. x_1 is sepal length, x_2 is sepal width, x_3 is petal length and x_4 is petal width. y_1 is *Setosa*, y_2 is *Versicolor* and y_3 is *Verginica*. **b** Experimental output intensity distribution of the IDNN for a class of flowers as “*Setosa*” is demonstrated. **c** Training and testing results of classification. One classification error appears on the testing part. **d** Confusion matrix in our experiment using 30 different flowers.

Comment 3: *Acronyms should be expanded in the abstract, IDNN integrated diffractive optical network.*

Answer 3: As pointed out by the reviewer, in the revised manuscript, the acronym IDNN is expanded in the abstract as

“The **integrated diffractive optical network (IDNN)** chip demonstrates a promising avenue towards scalable and low-power-consumption optical computational chips for optical-artificial-intelligence.” *in Line 14 on Page 2.*

Comment 4: *The meaning of the subscripts of E in Eq.(1), Eq.(S3), Eq.(S5) and Eq.(S6) should be specified and consistent.*

Answer 4: As suggested by the reviewer, E in Eq.(S3), Eq.(S5) and Eq.(S6) have been revised and are now consistent with the E in Eq. (1). Each E in these equations is defined explicitly in the revised manuscript as

In the main text: “ $E_i(k)$ is the normalized electric field after the ODFT operation.” *in Line 9 on Page 7.*

In the Supplementary material: “ $E_i(k)$ is the normalized electric field after the ODFT operation, $E_o(x_2)$ is the output field distribution of the array waveguide, and E_α is a distribution of eigenmodes of the output array waveguide.” *in Line 9 on Page 5.*

Comment 5: *There are 10 channels in the chip, however, the digit number of the input sequences in the caption of Fig.S9(b)(c)(d) is 11.*

Answer 5: As pointed out by the reviewer, the digit number in Figs. S7(b)-(d) has been corrected to 10 in the revised manuscript as

“**Fig. S7. The experimentally retrieved sequence with different input sequences. a** [1000000000]. **b** [0010000000]. **c** [1001000000]. **d** [1001001000].”

Comment 6: *Line 119: There is no Fig.S3a.*

Answer 6: As pointed out by the reviewer, “Fig. S3a” is corrected to “Fig. S5a” in the revised manuscript as

“All phase shifters are thermally tuned by the TiN heaters that are packaged to a printed circuit board (**Fig. S5a**).” *in Line 17 on Page 6.*

In summary, we have addressed all comments made by the reviewer. The manuscript and supplementary materials have been carefully corrected. We hope these answers and efforts could ensure a more complete and insightful manuscript, and we are also grateful for the Reviewer's essential contribution in instigating these improvements.

Reply to Reviewer 2

We are grateful to the Reviewer for the constructive comments and are delighted that the Reviewer is interested in our results. We are happy to address all the comments in line below. We thank the Reviewer for stimulating those improvements.

Comment 1: *The presented design has a clear advantage in scalability and power consumption compared to previous approaches. However, the manuscript lacks a critical discussion on the limitations of this approach. It should be noted that not all problems that can be solved with optical neural networks can be tackled with this approach.*

Answer 1: As suggested by the Reviewer, we have added a paragraph in the discussion section to claim the research scope of this approach, which are added in the revised manuscript as

“It is worth mentioning that the IDNN focuses on implementing the convolution matrix with the advantages of scalability and low power consumption, instead of an arbitrary matrix-vector multiplication modulation. Since a convolution matrix contains fewer free parameters than those in an arbitrary matrix, the current demonstrated one-layer network may not be an optimal solution for complicated classification tasks. This issue can be mitigated by implementing multi-layers or multiple convolutions in each layer to obtain a stronger expression capability. The speed of IDNN is also limited by the electrical equipment and can be accelerated when the modulators and detectors are high-speed programmed.” in Line 11 on Page 15.

Comment 2: *The authors mention that the training/learning of this photonic system for the machine learning classification tasks has been done using Pytorch. However, there is no mention to the number of parameters that are trained. There is also no mention to how the learned parameters are transferred from the numerical simulations to the experiments. For instance, it would interesting to know if the training has been informed with data coming from the experiments. The difficulty of a proper training of physical systems is a crucial aspect that appears to be disregarded in the current work.*

Answer 2.1: For the descriptions of parameter training and the transfer from simulation to experiment. We have added the details of the number of parameters that are trained in each task correspondingly in the revised manuscript as

“The input and output are connected by a complex, circulant weight matrix (W) with 8 trained complex-valued parameters.” in Line 22 on Page 9; and

“**Fig. 5a** shows that the network is composed of two layers, a hidden layer W with 16×10 trained complex parameters and an output layer W^{out} with 10 trained complex parameters.” in Line 11 on Page 11.

Answer 2.2: For the transfer of parameters from simulation to experiment.

Our training process is performed by simulation and the trained parameters are further transferred to the experiments. The transfer of parameters from the simulated model to the experiment is added in **Methods** in the revised manuscript as

“The training process of **Iris flower**, **Handwritten digit**, and **Fashion product** datasets was conducted in Pytorch (Google Inc.), a package of Python. The learned complex parameter (w) is expressed as $w = a + ib = Ae^{i\alpha}$ where A is encoded into the internal phase shifters (θ) with $\theta = \arcsin(A^2)$ and α is encoded into the external phase shifters (ϕ) with $\phi = \alpha - \theta/2$.” in Line 8 on Page 17.

Comment 3: *Along the main text, numerical and experimental results are reported. The separation between what has been estimated from numerical simulations and the results that have actually been obtained from experiments is often unclear. A more precise writing would benefit the manuscript.*

Answer 3: As pointed out by the reviewer, the contents to better distinguish between simulation and experimental results are added, in the revised manuscript as

1. “The **experimentally obtained** energy distribution from the eight output waveguides is shown in **Fig. 4b** for one testing data point as an example.” in Line 14 on Page 10.
2. “The one-layer IDNN chip can achieve a high classification accuracy of 96.7% over 30 testing images in the experiment (**Fig. 4d**).” in Line 23 on Page 10.
3. “Besides, the **simulated** energy distribution from the intensity detection results of 10 outputs is shown in **Fig. S11f** to display the classification performance, **indicating that our numerical simulations predict the channel with the highest energy will generally corresponds to the correct handwritten digit.** We experimentally tested 500 images and the confusion matrix (**Fig. 5c**) shows an accuracy of 91.4% in the generated predictions, in contrast to 92.6% for the numerical results obtained from simulations. The output energy distributions from 10 waveguides for the classification of the digit “2” are depicted in **Fig. 5d** (More experimental results are shown in **Fig. S12**).” in Line 16 on Page 11.
4. “The **experimental** energy distributions collected from 10 waveguide outputs using an image of a “pullover” is shown in **Fig. 5g** and the **experimental** results from other images are depicted in **Fig. S14**.” in Line 1 on Page 13.

Comment 4: *The manuscript contains several bold statements that I feel are too detached from reality. For instance, the sentence “...facilitating the realization of hundreds of millions of neurons” feels unrealistic given that here 10 input modes have been demonstrated. Similarly, the sentence “... can find tremendous applications...” should be more precise in terms of the applications that such a design can perform.*

Answer 4: As pointed out by the reviewer, we have removed the expression “facilitating the realization of hundreds of millions of neurons”. Meanwhile, we have extended our chip dimension to 128 modes using numerical simulations, showing an acceptable loss (**Fig. S1**).

For the sentence "... can find tremendous applications...", we have added specific applications and relevant references in the revised manuscript as

“Our integrated chip is adaptable to the Fourier-based convolution acceleration algorithm and multiple tasks, which has great potentials for **many** applications in compact and scalable application-specific optical computing, future optical-artificial-intelligence computers and quantum information processing, **such as image analysis, object classification, and live video processing in autonomous driving [5, 41].**” in Line 20 on Page 15.

Comment 5: *The performance for the classification tasks is relatively poor in the experimental evaluations. In particular, for the MNIST database, the authors report an accuracy of 89.3% in the experiments for 150 tested images. There are two aspects here that need further explanations. First, the authors should indicate how the 150 tested images in the experiment were extracted from the dataset of 10000 testing images. This is particularly relevant since some digits of the MNIST database are significantly easier to distinguish than others. Second, the performance for the MNIST of previous works in optical neural networks reach higher accuracies, close to 99%. See e.g. M. Hermans et al. Journal of Machine Learning Research 16 (2015) 2081-2097 or P. Antonik et al. IEEE Journal of Selected Topics in Quantum Electronics 26 (2020) 7700812. I wonder if the reduced performance here is due to the small size of the optical network or for the difficulty to transfer the training parameters from the numerical simulations to the experiments (please comment if there are other relevant factors).*

Answer 5.1: The extraction of testing samples. The 150 testing images in the experiment were uniformly and randomly extracted from the dataset of 10000 testing images. In the revised manuscript, we extend the sample number to 500, which is the same amount as in other references, such as *NATURE* 589.7840 (2021) (the Ref. 35). We add this sample extraction process in the revised manuscript as

“Our model is trained on the entire training set, and **500** instances **are uniformly and randomly drawn from the testing set** to validate the trained model on-chip [35].” in Line 9 on Page 11.

Answer 5.2: The reason for reduced performance. The reduced performance is due to the compression of input data and the limitation of the size of the optical network. In Ref. 50 and Ref. 51, they use photonic reservoir computing to perform the classification task. A reservoir is a fixed system, representing a ‘black-box’ with uncontrollable internal dynamics and thus is not updated during the training process. In this process, the **reservoir matrix** has many fixed nodes (around 200,000 nodes) using fiber delay or SLM, which leads to a high accuracy with 784 inputs. For our experiment, the dimensionality of the dataset is much higher than that of our optical neural chip. To reduce the computing burden and the number of input parameters, the image data is first compressed from the dimensionality of $28 \times 28 = 784$ to $4 \times 4 = 16$ inputs by converting the images to the k -space and extracting the low-frequency information (4×4

matrices in the center of the image). In this compression process, some information is lost, and thus lowering the classification accuracy. On the other hand, the number of neuron nodes is smaller in our designed network. Therefore, compared with previous works using the optical reservoir computing, our experiment has a lower classification performance. The main advantage of our design is its smaller footprint and lower energy consumption as compared to traditional fully connected networks. In the meantime, our approach has a comparable classification accuracy for some typical classification tasks [Refs. 7 and 13]. The description of the experimental results is added in the revised manuscript as

“We experimentally tested 500 images and the confusion matrix (Fig. 5c) shows an accuracy of 91.4% in the generated predictions, in contrast to 92.6% in the numerical simulation. The output energy distributions from 10 waveguides for the classification of the digit “2” are depicted in Fig. 5d (More experimental results are shown in Fig. S12). These results show that we have successfully implemented the classification task on the integrated diffractive optical computing platform. Our experimental testing accuracy is slightly lower (91.4%) than the numerically predicted value obtained from simulations (92.6%), which is attributed to the errors from the diffractive region, heater calibration and other experimental factors (Supplementary Note 10). Although our method does not achieve as high accuracy as previous reservoir computing demonstrations [50, 51], their internal dynamics of reservoirs are uncontrollable with fixed nodes, which leads to the stringent requirement of obtaining many parameters with high accuracy. Consequently, they are difficult to be implemented on chip. In contrast, our approach requires fewer parameters in controllable weight matrices that significantly reduces the footprint and energy consumption. Meanwhile, the input data compression process in our method also loses some information and lowers the numerical classification accuracy. The numerical blind testing accuracy (92.5%) with 10,000 samples can be further improved after enlarging the size of the network with more input information. It is worth noting that the classification accuracy obtained with our approach is comparable to the fully connected network (Fig. 5b), but our method consumes less energy and occupies less footprint, representing the major advantages of the design.” in Line 19 on Page 11.

The two references are added as

50. Hermans, M., *et al.* Photonic delay systems as machine learning implementations. (2015).

51. Antonik, P., Nicolas M., and Damien R. Large-scale spatiotemporal photonic reservoir computer for image classification. *IEEE J. Sel. Top. Quantum Electron*, 26.1, 1-12 (2019).

Comment 6: *The supplementary information is quite extensive and contains as many as 18 figures. In my opinion, it is unclear what is shown in several of these figures. I find that crucial descriptions and explanations are missing at several points of the text. Although this criticism affects mainly to the supplementary information, it partly affects to the results presented in the main text, which would also benefit from a more streamlined description of the results.*

Answer 6: As per the reviewer’s suggestion, we have revised the supplementary information by reducing the figures to 15, revising the captions of some figures [see page 4, 16, 19, 21, 23], and adding mathematical elucidation of correlation algorithms and the discussion about the Fourier-based convolution.

In summary, we have addressed all comments made by the reviewer. The limitations of this approach have been discussed and the experimental accuracy has been well analyzed. We hope these answers and efforts could ensure a more complete and insightful manuscript, and we are very grateful for the Reviewer's essential contribution in instigating these improvements.

Reply to Reviewer 3

We are grateful to the Reviewer for the constructive comments and the recognition of our design with higher integration for large-scale photonic computing. We are happy to address the comments. We thank the Reviewer for stimulating those improvements.

Comment 1: *The discussion section should make it clear that the proposed approach cannot implement the arbitrary matrix-vector multiplication but the convolution matrix. In the MZI-based ONN [17], the authors decomposed the learnable matrix in each neural network layer via SVD, and then implemented matrix multiplication with SU(4) core and DMMC core of performing unitary matrix multiplication and diagonal matrix multiplication, respectively. While in this proposed IDNN, the authors replaced the unitary matrix multiplication operations with two Fourier transforms via diffraction. The unitary matrix multiplication in [17] is learned from data, while the Fourier transform in IDNN is fixed. It is not surprising that substituting two learnable components with a fixed Fourier transform can reduce the footprint and energy consumption.*

Reply: As pointed out by the reviewer, our proposed approach cannot implement arbitrary matrix-vector multiplications as it has been optimized directly for the convolution matrix. We have added a paragraph in the discussion section to highlight this point, in the revised manuscript as

“It is worth mentioning that the IDNN focuses on implementing the convolution matrix with the advantages of scalability and low power consumption, instead of an arbitrary matrix-vector multiplication modulation. Since a convolution matrix contains fewer free parameters than those in an arbitrary matrix, the current demonstrated one-layer network may not be an optimal solution for complicated classification tasks. This issue can be mitigated by implementing multi-layers or multiple convolutions in each layer to obtain a stronger expression capability.” in Line 11 on Page 15.

Comment 2: *The experiment is conducted only on a small proportion of the testing dataset. For example, only 150 out of 10,000 different handwritten digits are tested in the MNIST dataset classification. What are the reasons that hinder testing all the digits in the testing set since both the MZI and detector can be high-speed programmed and reconfigured?*

Answer 2: We agree with the Reviewer to increase the number of the testing data points to be evaluated in the experiment. We have increased the testing sample size to 500, which is the same number as that used in the experiments in Reference [35] (*NATURE* 589.7840 (2021)). To facilitate the control of complex photonic devices, the driving equipment to manipulate the voltage of heaters is from *Qontrol Ltd.*, which has a modulation speed of ~1 kHz. For detection, the *NI-9215* with a sample rate of ~100 kHz is employed. In our experiment, the main factor to hinder the testing speed is our modulation speed of the electrical equipment (~1 kHz for the modulator). Nevertheless, the heaters in our chip could potentially reach tens of GHz when the thermo-optic modulators are replaced by high-speed carrier depletion modulators. When the MZI

modulators and detectors are high-speed programmed, the testing time of all digits in the experiment will be significantly reduced. This will be our next goal using high speed on-chip MZI modulators and detectors in our current chip. In this paper, we focus on the design of the computing chip to reduce the footprint and lower the energy consumption.

The experimental accuracies and related numerical results on 300 and 500 images for *MNIST* dataset are listed in the following table.

Table 1 The numerical and experimental results with amplitude encoding of *MNIST* dataset

Sample numbers	300	500	10,000
Numerical accuracy	91.7%	92.6%	92.5%
Experimental accuracy	90.0%	91.4%	---

As shown in **Table 1**, the process of selecting a small proportion of the testing dataset from 10,000 has a certain degree of randomness and the numerical accuracy may lower than 92.5%. But when the sample number extends to 500, the numerical accuracy is close to 92.5%. In addition, the relative errors between the experimental and the simulation results are kept within 2% for different sample numbers. Therefore, the sample size of 500 in experiment is sufficient to evaluate our design. The description of the testing accuracy is added in the revised manuscript as

“We experimentally tested 500 images and the confusion matrix (**Fig. 5c**) shows an accuracy of 91.4% in the generated predictions, in contrast to 92.6% in the numerical simulation. The output energy distributions from 10 waveguides for the classification of the digit “2” are depicted in **Fig. 5d** (More experimental results are shown in **Fig. S12**). These results show that we have successfully implemented the classification task on the integrated diffractive optical computing platform. Our experimental testing accuracy is slightly lower (91.4%) than the numerically predicted value obtained from simulations (92.6%), which is attributed to the errors from the diffractive region, heater calibration and other experimental factors (**Supplementary Note 10**). Although our method does not achieve as high accuracy as previous reservoir computing demonstrations [50, 51], their internal dynamics of reservoirs are uncontrollable with fixed nodes, which leads to the stringent requirement of obtaining many parameters with high accuracy. Consequently, they are difficult to be implemented on chip. In contrast, our approach requires fewer parameters in controllable weight matrices that significantly reduces the footprint and energy consumption. Meanwhile, the input data compression process in our method also loses some information and lowers the numerical classification accuracy. The numerical blind testing accuracy (92.5%) with 10,000 samples can be further improved after enlarging the size of the network with more input information. It is worth noting that the classification accuracy obtained with our approach is comparable to the fully connected network (**Fig. 5b**), but our method consumes less energy and occupies less footprint, representing the major advantages of the design.” in Line 19 on Page 11.

Similarly, the supplementary experiment results of *Fashion-MNIST* dataset are shown in Table 2. The relative errors between the experimental and the simulation results are kept within 3% for sample numbers of 300 and 500, showing that the experimental results agree well with the numerical estimations. We also revised and added some contents about the description of testing accuracy of *Fashion-MNIST* dataset in the revised manuscript as

“The corresponding accuracy for 500 testing images with one hidden layer is 82.0% in simulation versus 80.2% in experiment (**Fig. 5f**.)” in Line 22 on Page 12.

Table 2 The numerical and experimental results with phase encoding of *Fashion-MNIST* dataset

Sample numbers	300	500	10,000
Numerical accuracy	81.3%	82.0%	81.7%
Experimental accuracy	79.0%	80.2%	---

Figure 5 is also revised by replacing the confusion matrices of 150 samples from **Fig. 5c** and **Fig. 5f** with the experimental sampling results of 500 samples.

Fig. 5 Handwriting and Fashion recognition using the IDNN. a The network consists of an hidden layer W and a output layer W^{out} . The outputs of the output layer are recognition results. The input layer is calculated by the traditional computer and the complex output results of the input layer are converted into amplitude and phase information as the input of the chip. **b** The numerical testing results of accuracy and loss versus epoch number for the *MNIST* dataset. **c** The confusion matrix for our experimental results, using 500 different handwritten digits. **d** The

output intensity distribution of the IDNN for a handwritten input of “2” is demonstrated. **e** The numerical testing results of accuracy and loss versus epoch number for the *MNIST-Fashion* dataset. **f** The confusion matrix in the experiment. **g** As an example, the output intensity distribution of the IDNN for a fashion product input of “pullover” is demonstrated.

Comment 3: In Fig. 3, the digits of 4 and 9 are highly similar, resulting in the large peak values in the non-diagonal line. I would suggest using the different digits to make the statement in the last paragraph of page 8 more valid.

Answer 3: As per the reviewer’s suggestion, we have changed the digit 4 to 2 to avoid the large central peak value in the non-diagonal line. Previous results have been moved to **Supplementary Note 6**. The related experimental results are shown in **Fig. 3c**.

Fig. 3 Image recognition. **a** Schematics of the image recognition process. The input test image dimension is $n \times n$, the target image dimension is $m \times m$ and the output dimension is $k=n+m-1$. **b** Experimental normalized intensity from the output waveguides when the sample sequence [10101] is the same as the target sequence [10101]. **c** Experimental correlation results for digit image recognition. There are three-digit images (1, 2, 9), and each digit image is composed of a 5×5 matrix.

Comment 4: The manuscript compares the optical convolution networks and electronic fully-connected networks to prove the performance of their suggested structure. I also suggest the comparisons and discussions with the electronic convolution network as well as the state-of-the-

art research work of photonic convolutional accelerators, e.g., (1) Xu, X., et al. "11 TeraFLOPs per second photonic convolutional accelerator for deep learning optical neural networks." *Nature* 2021; and (2) Feldmann, J., et al. "Parallel convolutional processing using an integrated photonic tensor core." *Nature* 2021. Implementing the convolution in Fourier-space is more efficient for the larger kernel size but not for the small kernel size often used in CNN.

Answer 4: As per the reviewer's suggestion, we discussed the comparisons with the electronic convolution network and the state-of-the-art research work of photonic convolutional accelerators, in the revised manuscript as

"Our approach to convolutional processing provides an effective method for accelerating computation via reducing computational complexity, as compared with traditional methods [42-43]. The traditional electronic single convolution operation involves $(n - m + 1)^2$ matrix-vector multiplication (MVM) operations with $n \times n$ and $m \times m$ dimension matrices [42, 43]. Similarly, the state-of-the-art research works for photonic convolutional accelerators [5, 35] also perform convolution operation by MVM operations, but in combination with parallel operations using WDM technology to accelerate the computing. Here, we realize an optical Fourier transform-based convolution. Each 2D convolution with $n \times n$ and $m \times m$ dimension matrices needs $(n - m + 1) \times m$ 1D convolution operations [46]. Since a 1D convolution is converted into an MVM operation in our design, our approach only needs $(n - m + 1) \times m$ MVM operations to compute a 2D convolution. This design can thus accelerate the convolution operations to achieve improved scaling over traditional electronic counterparts [42, 43]. More discussions on the computational convolution approach can be found in **Supplementary Note 6**. It can further be extended to circulant convolution operation [45] and employed in typical classification tasks [47-49]." *in Line 3 on Page 9.*

In **Supplementary Note 6**, we added a paragraph to discuss the Fourier-based convolution in CNN as

"We further discuss the issue of applying the Fourier-based convolution in CNN. For the classical electronic network, the implementation of convolution in Fourier-space has no distinct acceleration advantage for small kernel size that is often used in CNN. As we know, the Fourier-based convolution has three main computing parts: Fourier transform, element multiplication, and inverse Fourier transform. In the optical field, as Fourier transform and inverse Fourier transform can be implemented passively without resource consumption, Fourier-based convolution can still realize a computational acceleration for the small kernel size used in CNN." *in Line 12 on Page 14.*

Comment 5: *The authors attribute the larger error between the experimental and the simulation results on MNIST than on FASHION MNIST to the error of amplitude modulation. Can the authors add a phase-only modulation experiment on MNIST to prove this suspect? Or will the exclusion of amplitude modulation lead to obvious accuracy loss on MNIST?*

Answer 5: As pointed out by the reviewer, the larger error may or may not be attributed to the error of amplitude modulation according to our experimental results. Therefore, as per the reviewer's suggestion, we have further added an experiment with phase-only encoded input

signal on *MNIST* and an amplitude encoded experiment on *Fashion-MNIST* with 300 and 500 samples, respectively, as shown in **Table 3** and **Table 4**.

Table 3 The numerical and experimental results with phase encoding of *MNIST* dataset

Sample numbers	300	500	10,000
Numerical accuracy	92.0%	92.6%	92.4%
Experimental accuracy	91.3%	89.4%	---

Table 4 The numerical and experimental results with amplitude encoding of *Fashion-MNIST* dataset

Sample numbers	300	500	10,000
Numerical accuracy	81.3%	81.4%	81.4%
Experimental accuracy	79.0%	80.4%	---

For the *MNIST* dataset, the experimental accuracy under phase encoding is slightly lower than that under the amplitude encoding. While for *Fashion-MNIST* dataset, the experimental accuracy under phase encoding is almost the same as that under the amplitude encoding. Therefore, we remove the claim that amplitude encoding leads to obvious accuracy loss. At the same time, the experimental results are added into the Supplementary material **Fig. S15**. The description on the error analysis is also revised as

“We implement additional experiments using the phase channel encoding input signal on *MNIST* dataset and using the amplitude channel encoding input signal on *Fashion-MNIST* dataset. The confusion matrix for 500 images (**Fig. S15a**) shows an experimental accuracy of 89.4%, in contrast to 92.6% for the numerical simulation results on the *MNIST* dataset. The results for *Fashion-MNIST* dataset are 81.4% according to the numerical simulations versus 80.4% for the experiment (**Fig. S15b**). We note that the accuracy error between the experimental and numerical results has no distinct difference between using the amplitude channel and the phase channel. However, for phase encoding, the intensity of the input signal would not be attenuated, which leads to a higher output intensity in our experiment.” in Line 3 on Page 13.

Fig. S15 The confusion matrix for experimental results using 500 different samples with **a** phase encoding of *MNIST* dataset; and **b** amplitude encoding of *Fashion-MNIST* dataset.

In summary, we have addressed all comments made by the reviewer. More extensive experimental testing on 300 and 500 images for *MNIST* dataset and *Fashion-MNIST* dataset with phase-only and amplitude encoded input signal are added. We hope these answers and efforts could ensure a more complete and insightful manuscript, and we are very grateful for the Reviewer's essential contribution in instigating these improvements.

REVIEWERS' COMMENTS

Reviewer #1 (Remarks to the Author):

We went through the revised manuscript, the reports from other referees, and the author's replies. In this round, the authors have made considerable improvement on their manuscripts (especially, added discussion on correlation pattern recognition and Hadamard product operation, the possible limitations of the proposed approach, etc.) and addressed all the comments/suggestions of the referees. We are satisfied with their responses and efforts with regard to our comments/suggestions. We are happy to recommend its acceptance in Nature Communications with the current version.

Reviewer #2 (Remarks to the Author):

The authors have improved the manuscript after the revisions.

The methodology has been clarified and the quality of the results has been improved by increasing the size of the analyzed dataset.

The authors are also more precise in the comparison with the state-of-the-art and the potential of the presented approach.

I recommend publication of the manuscript in Nature Communications.

Reviewer #3 (Remarks to the Author):

The authors have provided sufficient revisions and discussions to address my comments. I could understand the potential value of this work, and I don't have any other reasons against the publication of this work in Nature Communication.